# Generation of Highly Charged Au Ion in Laser-Produced Plasma for Water Window X-ray Radiation Sources

**Jiahao Wang [1], Maki Kishimoto [1], Tomoyuki Johzaki [1], Kairi Mizushima [1], Chihiro Kumeda [1], Takeshi Higashiguchi [2], Atsushi Sunahara [3], Hikari Ohiro [1], Kotaro Yamasaki [1] and Shinichi Namba [1,\***

1   Department of Advanced Science and Engineering, Hiroshima University, Hiroshima 739-8527, Japan
2   Department of Electrical and Electronic Engineering, Utsunomiya University, Utsunomiya 321-8505, Japan
3   Center for Material under Extreme Environment, Purdue University, West Lafayette, IN 47907, USA
*   Correspondence: namba@hiroshima-u.ac.jp; Tel.: +81824247615

**Abstract:** Highly charged ions in the plasma produced by high-power laser can radiate bright and short-pulse X-rays. Owing to the unresolved transition array (UTA) from the high-Z elements, laser produced plasma has been applied for developing X-ray sources. In particular, X-rays in the water-window (WW) region (2.3–4.4 nm) is utilized as the light source of the X-ray microscopy to observe living organisms under high contrast and resolution. In this work, WW X-rays radiated from a laser (1064 nm, 6.2 ns) produced Au-plasma has been studied. UTA spectrum in the WW range has been observed through a grazing incident spectrometer (GIS). Dependence of Au-ion charge state distribution on laser intensity has been experimentally investigated and evaluated by a transition probability data calculated by the flexible atomic code. The integrated soft X-ray emission has been observed through a pinhole camera with a 1.0-μm Ti-filter, combined with a 2-D plasma radiation scanning achieved by the GIS. An intense WW emission region 200-μm away from the target surface has been observed, which indicates a more effective area is possible to be utilized for a practical use.

**Keywords:** laser produced plasma; water window X-rays; Au highly charged ions

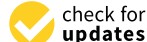



## 1. Introduction

Highly charged ion (HCI) produced from high-Z elements is essential for research on astronomy, astrophysics, and nuclear physics [1,2]. Applications using its features, such as laser spectroscopy and atomic clock, have been improved in the past decades [3–5]. On the other hand, transitions from the countless energy levels in the HCI emit continuous spectral radiation, making it suitable for developing short-wavelength light source when compared with line emissions from low-Z elements [6,7]. Particularly, soft X-ray microscopy (SXM) using a light source in the so-called water-window (WW) region (wavelength 2.3–4.4 nm, photon energy 281–540 eV) has been considered an important tool for observing living organism cells with a several-nanometer-scale resolution [8,9]. The WW region is defined as the transmission window between the K-absorption edges of carbon and oxygen, where oxygen atoms are almost transparent to the WW X-rays, while carbon has high absorptivity to them. Consequently, high contrast imaging can be obtained from living biological cells under SXM [10].

Since the photon flux, which influences the sample exposure time, is a crucial parameter when evaluating a WW light source, several setups have been attempted to reach a sufficient WW X-rays output [11]. Radiation sources utilizing synchrotron can generate bright and fs-scale WW pulse [12], however, their cost and large equipment highly constrains small and mid-research facilities to deploy them practically. In contrast, s WW source using plasma produced by table-top laser is more favorable in size and cost. The tight-focused laser spot can heat the plasma effectively to generate HCIs [13,14].

Optics for soft X-ray (SXR) regions are also limited due to the poor reflectivity of the mirrors [15,16]; however, since a contact imaging method the whole band of the WW can

contribute to the sample exposure, the UTA emission emitted from the HCI is then suitable for SXR development. According to a Quasi-Moseley's law reported by [17], elements with an atomic number near 79 generates UTA emissions with their peaks in the WW region, which both *n* = 4-4 and *n* = 4-5 transitions from HCI contributes to them. However, a more detailed optimization is required where specific HCI for WW emission can be generated properly, which is possible to enhance the conversion efficiency (CE) of the WW source. In this work, dependence of the WW X-rays emitted from the laser-produced Au-plasma in laser energy is investigated by using a commercial table-top Nd:YAG laser. Combined with calculation results achieved by using the flexible atomic code (FAC) [18], a correlation between the highly charged Au ions and the incident laser power has been clarified. Additionally, laser-produced Au plasma is spatially studied to find the most intensive WW radiation area in the plasma plume.

## 2. Experimental Setup

The laser pulse for plasma inducing is generated from a table-top Nd:YAG laser system (wavelength:1064 nm, output energy: <1.2 J). Several procedures are set to maintain and monitor the laser pulse in a stable condition. The seeder-injected laser works in a 10 Hz repetition rate with a ~6.2-ns pulse duration. A high-speed digital shutter is placed in the beam line, cutting out 1 pulse to pass through an optical isolator. Since both the plasma surface that reaches the critical density of the laser, $\sim 10^{21}$ cm$^{-3}$, and the Au foil surface can reflect the incoming laser, the optical isolator can prevent the reflected light from damaging the laser optics. An energy-pulse monitor consists of a calorimeter, a fast Si photodiode (raise time 700 ps), and two glass plates set in the beamline before the chamber to monitor the laser energy and pulse shape shot-by-shot to reduce the influence of the laser fluctuation. Figure 1 shows a schematic of the experimental chamber and part of the beam line. The laser is focused through a lens ($f = 100$ mm) onto an Au target, which is mounted on motorized stages. A tightly focused spot with a full width at half maximum (FWHM), 15 μm, has been observed, reaching an $\sim 6.4 \times 10^{13}$ W/cm$^2$ on target. When a laser hits the target surface, the laser ablation processes can carry lots of solid debris (atom, molecule, nanoparticle, and microparticle), which can bring damage to the optical components. Therefore, a debris shield (thin glass plate) is placed behind the focus lens.

The spectral emission from the plasma is observed by a flat-field grazing incident spectrometer (grating 2400 grooves/mm) with a toroidal mirror, placed perpendicular to the incident laser. A back illuminated type X-ray charge-coupled device (X-CCD) is employed to measure the temporally integrated spectra. Wavelength calibration is achieved by using the carbon H-like and He-like resonance lines. On the other side, a pinhole camera (pinhole size 25 μm) is installed at 45 degrees with respect to the target surface, observing the temporally integrated SXR emission through a 1-μm-thick Ti-filter.

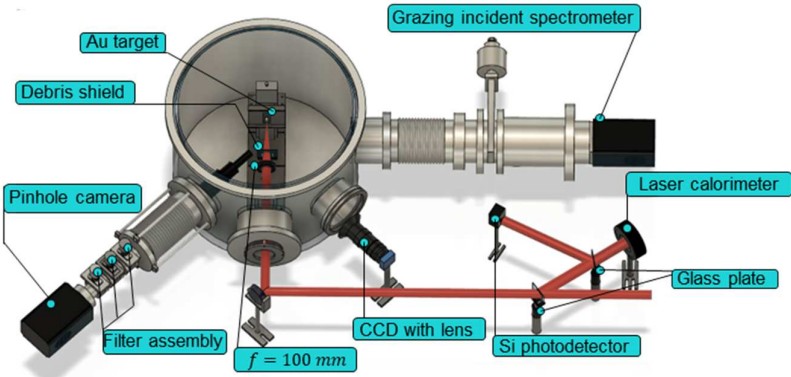

**Figure 1.** Schematic of the experimental set up (laser beam line, target system, and detectors).

## 3. Results and Discussions

The SXR emissions measured by the pinhole camera under different input laser energies are shown in Figure 2. Emission length and height have been rescaled when considering the plasma plume is simply 45 degrees to the pinhole. Noting that the emission region in Figure 2 does not only contain WW X-rays, it also contains out-of-band SXRs (wavelength shorter than 2.3 nm or from 4.4 nm to 10 nm) that can also pass through the 1-μm Ti filter according to a pinhole camera sensitivity calibration. The contour lines in the Figure 2 represent the half maximum of the intensity where hot and dense plasma is generated. Since the laser pulse has a nanosecond duration, both laser-target and laser-plasma interactions should be considered. After the laser pulse hits the target, an initial plasma is formed, and Au ions suffer from collisional heating while the rest of the laser pulse radiates the plasma. As the temperature increased, electron density of the central area reaches the critical density, causing the laser beam can no longer penetrate the plasma. Thus, hot and dense plasma remains around the critical density region, emitting strong SXR emissions including photons over 1 KeV, which contribute to the red regions in Figure 2. By contrast, SXRs emitted from the expanded plasma contributes to the green and blue regions in Figure 2. The FWHM obtained from the cross section of the vertical and horizontal directions has also been shown in the figures.

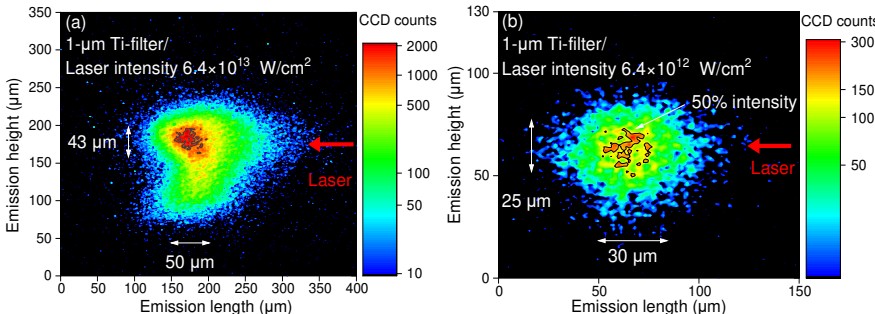

**Figure 2.** Pinhole images with 1-μm Ti filter for laser intensities of (**a**) $6.4 \times 10^{13}$ and (**b**) $6.4 \times 10^{12}$ W/cm$^2$. The laser pulse emits from the right side is shown in red.

SXR spectra radiated from the Au-plasma is measured by the GIS with a slit width of 350 μm. Although spatial resolution of the GIS gets worse as the slit width increases, a wider slit interval enables GIS to measure X-rays emitted from a broader region of the plasma. The laser produced Au spectra in the WW region at $6.4 \times 10^{13}$ W/cm$^2$ is shown in Figure 3a. The peaks, which mainly contributes to each ionic charge state, are illustrated by the arrows. According to the spectra, Au$^{27+}$ ion is dominated in the laser plasma, producing intense WW X-ray emissions at around 2.6 nm.

According to a previous laser-plasma simulation by using the STAR2D, Au$^{28+}$ ions are dominated in the dense plasma with an open shell of four in this case [19]. Transition possibility of 4d-4f and 4f-5g transitions from Au$^{20+}$ to Au$^{30+}$ ions are then calculated by the FAC. The calculation results are shown in Figure 3b, compared with intensity-normalized Au spectrum for different laser energies. Spectral peak shifts from 2.6 nm to 3.0 nm are observed as the laser energy reduces, which mainly contributes to the *4f*-*5g* transitions from Au$^{27+}$ to Au$^{24+}$ ions. Au ions with higher charge states become dominated in the plasma when the electron temperature gets higher due to a high laser intensity, resulting in the peak of UTA contributes to the *4f*-*5g* transition shifts as the calculation results illustrates. Nevertheless, CCD counts beyond 4.2 nm contributed to the n = 4-4 transitions show similar peaks at around 4.5 nm, which is out of the WW band.

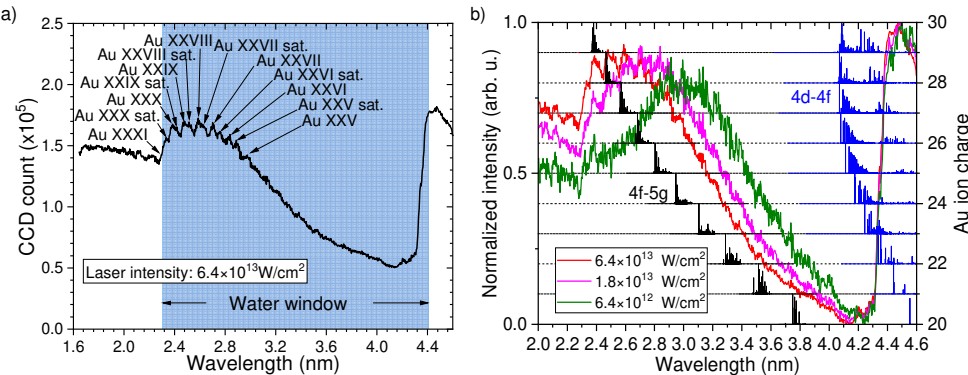

**Figure 3.** (**a**) Au spectra in WW region. (**b**) Au spectrum under different laser intensities (red curve: $6.4 \times 10^{13}$ W/cm$^2$, pink: $1.8 \times 10^{13}$ W/cm$^2$, green line: $6.4 \times 10^{12}$ W/cm$^2$,), with a comparation of the FAC calculation results. Emission wavelength with normalized transition probabilities (gA) of $4f$-$5g$ (black lines) and $4d$-$4f$ (blue lines) transitions in Au$^{20+}$-Au$^{30+}$ is shown in each row.

To obtain a spatially spectral emission distribution, a 2D spectra scanning along the laser incident direction is conducted by utilizing the GIS with a 50-μm slit width (shown in Figure 4). A 30-μm Au foil is irradiated by a pulse energy of $6.4 \times 10^{13}$ W/cm$^2$ and the surface of the target refers to the observation position of 0 μm. A peak CCD count at the observation position of 200 μm has been observed, where both $n$ = 4-4 and $n$ = 4-5 transitions emit intense UTA. Noting this, the difference in emission size between the pinhole camera image and the GIS result is caused by the same spectral range cannot be observed by two measurements due to Ti filter transmission curve.

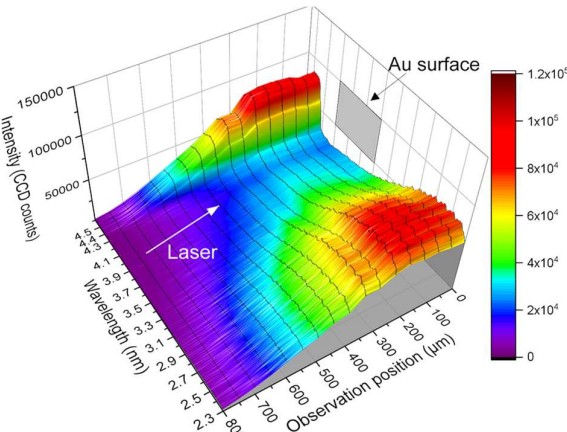

**Figure 4.** The 2-D spectral emission behavior irradiated with a laser intensity of $6.4 \times 10^{13}$ W/cm$^2$. The position where GIS observed on the Au surface is calibrated to the observation position of 0 μm.

## 4. Conclusions

WW emissions from highly charged Au-ions produced by a nanosecond laser has been spatially and spectrally studied. The dependence of the Au ions on laser energy has been investigated by changing the incident laser power and using the FAC calculations. The dominated Au-HCI in the laser-produced plasma dropped from Au$^{27+}$ to Au$^{24+}$ as the laser power changed from $10^{13}$ W/cm$^2$ to $10^{12}$ W/cm$^2$. A broad UTA spectral band, which is mainly contributed by the Au 4f-5g transitions, produces abundant X-rays in the WW region, which can be fully utilized for the WW soft X-ray microscopy. The 2-D emission spectra scanning from the Au surface has also been measured, and an intense WW region 200-μm away from the target surface is shown to be suitable for setting the observing sample when considering a contact-type X-ray microscopy. A comprehensive understanding of

the Au-plasma can produce valuable optimizations for WW X-ray microscopy to be more practically used in the future.

**Author Contributions:** Conceptualization, M.K., T.H. and S.N.; Methodology, J.W., M.K., K.Y. and S.N.; Software, J.W., T.J., C.K., T.H. and A.S.; Formal analysis, J.W.; Investigation, J.W., T.J., K.M., H.O., K.Y. and S.N.; Resources, J.W., M.K., T.J., T.H., A.S., K.Y. and S.N.; Data curation, J.W., M.K., K.M. and A.S.; Writing—original draft, J.W.; Writing—review & editing, S.N.; Supervision, S.N.; Project administration, J.W. All authors have contributed to the manuscript equally. All authors have read and agreed to the published version of the manuscript.

**Funding:** This research was supported by JSPS KAKENHI under Grant No. JP20H00141, JKA Foundation (KEIRIN RACE).

**Institutional Review Board Statement:** Not applicable.

**Informed Consent Statement:** Not applicable.

**Data Availability Statement:** Not applicable.

**Acknowledgments:** This work was partially supported by Japan/U. S. Cooperation in Fusion Research and Development.

**Conflicts of Interest:** The authors declare no conflict of interest.

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
