# Peer review of "Generation of Highly Charged Au Ion in Laser-Produced Plasma for Water Window X-ray Radiation Sources"

_atoms, doi:10.3390/atoms10040150_

Round 1
Reviewer 1 Report
This paper describes the soft X-ray emission in the so-called water-window wavelength region from laser-produced gold plasmas produced by irradiating an Nd:YAG laser focused onto a gold target. Soft X-ray images are taken by a pinhole camera, while the emission spectra are measured by a grazing incident spectrometer. The authors also measure spatially resolved spectra to investigate the spatial distribution of the soft X-ray emission.
Though the analysis of a similar spectra has already been reported in ref. [10], soft X-ray images and spatially resolved spectra have been newly reported in this paper.
The reviewer judges that the contents of the paper should be worth publishing. However, some ambiguous expressions in Figure 2 should be clarified before acceptance:
Figure 2: The arrangement of the target is not shown in the figure. Where is the target and which direction is the normal to the target surface? This should be clarified.
The lengths of the vertical/horizontal arrows drawn in the figures seem to disagree with the scales of the corresponding axes. In addition, the arrows in (a) look clearly longer than the FWHM.
The reviewer also suggests some minor corrections of grammatical errors to clarify the descriptions as listed below:
line 20: ... by the GIS, an intense ... -> ... by the GIS. An intense ...
line 32: WW -> water-window (WW)
line 69: ... plates is set ... -> ... plates set ...
line 92: ... in Fig.2 not only contains WW X-rays, but SXRs ... -> ... in Fig.2 does not only contain WW X-rays, SXRs ...
line 102: contributes -> contribute
line 143: in laser energy -> on laser energy
line 147: which mainly ... -> which is mainly ...
line 149: ... measured, an ... -> ... measured, and an ...
Author Response
Thank you very much for reviewing our paper. We greatly appreciated your valuable comments and suggestions.
According to them, we have revised the paper, and herewith submit the revised version for your review.
Our responses to the comments/suggestions are writen in the word file. Please check more details in the attachment.

Reviewer 2 Report
I was carefully read the paper “Generation of highly charged Au ion in laser-produced plasma for water window X-ray radiation sources” by Jiahao Wang et al. and I think that it contains interesting results, and it can be accepted for publication. The paper is clear and well organized, and it could be of real interest for researchers in the field of laser-produced plasma.
Before the paper is accepted, I only have one comment that the author needs to respond to: The author identified that there are 4d-4f and 4d-5g transitions of Au20+- Au30+ions in the water window region. The author needs to provide more evidence or further comments to explain the accuracy of the results such as energy levels and transition wavelengths.
Author Response
Thank you very much for your kind comments. Please check the attachment for details of our respondse.
